# RECAL: Sample-Relation Guided Confidence Calibration over Tabular Data

**Haotian Wang**[1]    **Zhen Zhang**[1]    **Mengting Hu**[1*]    **Qichao Wang**[2]    **Liang Chen**[2]
**Yatao Bian**[3]    **Bingzhe Wu**[3]

[1] College of Software, Nankai University

[2] School of Computer Science and Engineering, Sun Yat-Sen University, [3] Tencent AI Lab

wanght@mail.nankai.edu.cn, mthu@nankai.edu.cn

## Abstract

Tabular-format data is widely adopted in various real-world applications. Various machine learning models have achieved remarkable success in both industrial applications and data-science competitions. Despite these successes, most current machine learning methods for tabular data lack accurate confidence estimation, which is needed by some high-risk sensitive applications such as credit modeling and financial fraud detection. In this paper, we study the confidence estimation of machine learning models applied to tabular data. The key finding of our paper is that a real-world tabular dataset typically contains implicit sample relations, and this can further help to obtain a more accurate estimation. To this end, we introduce a general post-training confidence calibration framework named RECAL to calibrate the predictive confidence of current machine learning models by employing graph neural networks to model the relations between different samples. We perform extensive experiments on tabular datasets with both implicit and explicit graph structures and show that RECAL can significantly improve the calibration quality compared to the conventional method without considering the sample relations.

## 1 Introduction

Tabular-format data widely exists in various real-world applications such as healthcare (Avati et al., 2020), credit modeling (Clements et al., 2020), and finical fraud detection (Ngai et al., 2011). Various machine learning models (Huang et al., 2020; Chen et al., 2019) are proposed for modeling tabular-format data, ranging from the traditional model such as GBDT (Friedman, 2001) to advanced deep models such as Transformer (Vaswani et al., 2017). These methods have achieved remarkable progress in both industrial applications (Fu et al., 2019) and data science competitions (Vanschoren et al., 2014).

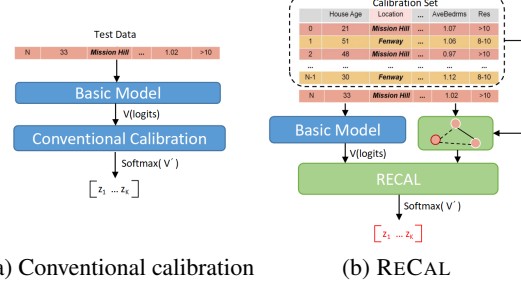

Figure 1: While conventional calibration methods focus only on the test sample itself, RECAL focuses on the relations between different samples.

Despite these progresses, the reliability of current models trained on tabular data still remains largely unexplored. Specifically, the predictive confidence of the model is the key aspect of reliability, which measures the probability that predictions will be correct. A high-quality confidence estimator is desired to deploy these ML models to risk-sensitive scenarios such as finance and healthcare (Amodei et al., 2016). A natural way to construct the predictive confidence is based on the maximum value of the output probability vector (Guo et al., 2017). Unfortunately, current ML models are prone to output inaccurate predictive distributions, which further lead low-quality confidence estimation. For example, as prior work shows (Kumar et al., 2018), Transformer models typically produce over-confident predictions (Mao et al., 2021). Besides, even for traditional ML models such as GBDT and random forests, we also show that there is room for improvement in their predictive confidence quality.

To provide accurate predictive confidence estimation of ML models, numerous post-hoc calibration techniques are presented for improving the confidence estimation in the post-training stage without modifying the original training procedure (Platt et al., 1999; Wang et al., 2021). The core idea of

---

*Corresponding author.

these techniques is to transform the original predictive probability into the calibrated one with some learnable transformation parameters. One common method is temperature scaling (Guo et al., 2017), which has been widely adopted for calibrating both deep and traditional ML models (Huang et al., 2017; Abdar et al., 2021). However, most of these methods are designed for general settings without considering the unique properties of tabular-data format.

This paper aims to improve previous calibration techniques over tabular-format data by considering the implicit relations between different samples. The motivation behind our work is based on the natural observation that real-world data tables contain implicit sample relationships, which can often be reconstructed by mining feature similarities. Figure 1 shows a real-world tabular dataset used for house price forecasting. In general, the prices of houses with similar latitude and longitude are always correlated. In other words, home prices are more closely related to their location. These sample relationships can be further used to improve the model confidence. Based on this observation, we introduce a general calibration method for ML models — RECAL, which provides a confidence calibration model with the ability to aggregate potentially correlated data across the table as clues for calibration. To integrate the implicit sample relation with the confidence, we build an undirected graph on top of them, where nodes are the samples in the table and edges represent the connections between them. A Graph Neural Network (GNN) is employed to aggregate logits from different samples, calibrating confidence. RECAL learns a unique temperature $t$ for each sample for scaling, hence retaining the accuracy of the basic model. By incorporating these implicit sample relations into the calibration process, we empirically show that the quality of estimated confidence has been consistently improved. Moreover, RECAL demonstrates remarkable robustness and stability against data noise and distribution shift. In a summary, the main contribution of this paper is as follows:

- To the best of our knowledge, we are the first to explore how to leverage the sample relation to improve the model confidence calibration over the tabular dataset.

- For the above problem, we present a GNN-based calibration framework named RECAL

- We demonstrate the effectiveness of RECAL on a wide range of ML models and tabular datasets with both implicit and explicit graph structures.

## 2 Related Work

**Graph Neural Networks.** Graph Neural Networks (GNNs) learn latent representations of node v through graph structure and features of nodes for tasks such as regression and classification (Kipf and Welling, 2016). It performs well when node features have homophily properties (Wu et al., 2020). Graph Convolutional Networ (GCN) is an important branch of GNN. Referring to CNNs, modern GCNs are based on the spectrum of graph Laplacian and learn local and global structural patterns of graphs by designing convolution and readout functions (Bruna et al., 2014). Current popular GCN networks typically use a neighborhood aggregation approach, also known as a message-passing mechanism, in which each node's representation is represented by a nonlinear aggregation function applied to its neighbors representation (Fey and Lenssen, 2019). A GCN model usually consists of multiple layers, each representing a nonlinear message-passing function to aggregate the local neighborhood of nodes.

**Confidence Calibration.** Confidence Calibration has been gaining more and more attention recently. Given a class prediction $\hat{Y}$ and a data set X, the model being perfectly calibrated should satisfy:

$$\mathbb{P}(\hat{Y} = Y \mid \mathbb{P}(\hat{Y} \mid X) = p) = p, \forall p \in [0, 1]. \quad (1)$$

Thus, confidence calibration changes the predictive distribution (also called logits) of the original model. For the tree-based model, Malinin et al. (2021) uses Stochastic Gradient Boosting (SGB) or Stochastic Gradient Langevin Boosting (SGLB) to generate ensembles of GBDT models. NGBoost (Duan et al., 2020) is based on natural gradients and uses the parameters of the conditional distribution as the target for Multi-parameter boosting. All of the above methods require to retraining the basic model from scratch. In deep learning, Order-Preserving Functions (Rahimi et al., 2020) train the calibration function on the basic models using a post-hoc method through a held-out dataset. Scaling is the commonly used post-hoc confidence calibration technique. On the binary model, Platt scal-

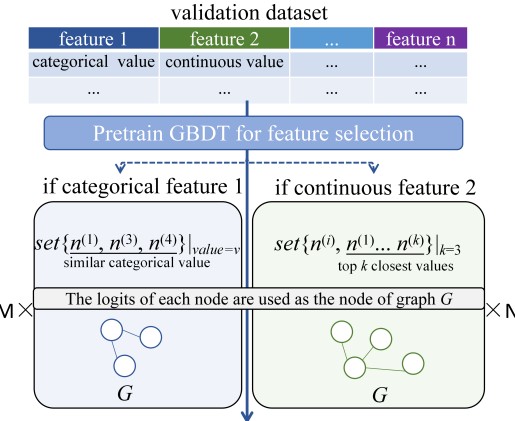

Figure 2: The process of constructing a graph from a table. A GBDT model is pre-trained using the validation set to select the feature with the largest contribution, and nodes are divided into different sets based on the selected feature. Edges exist between nodes in the same set. M indicates that the categorical feature has M different possible values. N indicates the total number of nodes.

ing (Platt et al., 1999) transforms logits by predicting two scalar parameters. Temperature scaling and Matrix scaling (Guo et al., 2017) are multiclassification extensions of Platt scaling. CaGCN (Wang et al., 2021) uses neural networks as nonlinear post-hoc calibration functions. However, how to find implicit sample relations from the tabular data and construct a generic calibration model for tabular data independent of the basic model remains to be studied.

## 3 Methodology

In this section, we introduce a post-hoc calibration method called RECAL to calibrate the model applied to the tabular data. Assume in a tabular classification task, we have a basic model $\mathcal{M}$, which can classify each row into $K$ classes. Yet $\mathcal{M}$ is not well-calibrated, indicating its confidence cannot accurately reflect its performance. Our purpose is post-calibration, which indicates that the training data is unavailable at this stage. Only a small held-out dataset can be used (Guo et al., 2017). Therefore, we aim to train another calibration model, i.e. RECAL $\mathcal{C}$, with the held-out dataset $\mathcal{R} = \{r_1, r_2, ..., r_N\}$, where $N$ indicates the number of samples in $\mathcal{R}$. RECAL achieves calibration for each sample by leveraging the graph structure built from implicit sample relations.

Specifically, as depicted in Figure 3, RECAL first pre-trains a feature selection model and constructs

a graph structure implicit in the table based on the selected feature (§3.1). Then we use GNN as a non-linear calibration function to characterize the topology on the graph (§3.2). Finally RECAL outputs a unique temperature $t$ for each sample and calibrates samples using a scaling-based approach (§3.3).

### 3.1 Sample-Relation Modeling

Tabular datasets are widely used in machine learning models for training and inference purposes. They exhibit a highly structured format, where each column represents a specific feature or variable, and the values in different rows of the same column are treated as independent entities. However, it is important to acknowledge that each column often carries its own physical significance, and certain features may have a more intuitive impact on the predictive outcomes. For instance, in the context of *house price forecasting*, houses located in *similar geographic regions* tend to exhibit similar prices. Hence, it is hypothesized that implicit relations exist between samples in the dataset, which could potentially enhance the confidence and accuracy of predictions. Motivated by this, we model the sample relations in the held-out dataset to build a graph. The process is depicted in Figure 2.

**Pre-train GBDT**  Specifically, we pre-train a Gradient Boosting Decision Tree (GBDT) model using the validation set in the dataset. The utilization of the validation set is motivated by the consideration of post-hoc calibration scenarios (Guo et al., 2017). *In this way, the original training data and the basic model $\mathcal{M}$ remain black boxes during the calibration process, thus ensuring user privacy protection.* Additionally, the adoption of GBDT is appropriate due to its rapid convergence, low training overhead, and its efficacy as an auxiliary decision-maker. Moreover, GBDT exhibits high interpretability, accurately expressing the importance of different features (Friedman, 2001).

**Graph Construction**  Based on the good explainability of GBDT, we select the feature that contributes the most significantly to the GBDT results as the foundation for our graph construction. If the selected feature is categorical, we establish connections between nodes sharing the same categorical value for that feature. For continuous features, each sample is connected to the top-$k$ samples that demonstrate proximity. By incorporating these implicit relations into the model, our objective is to

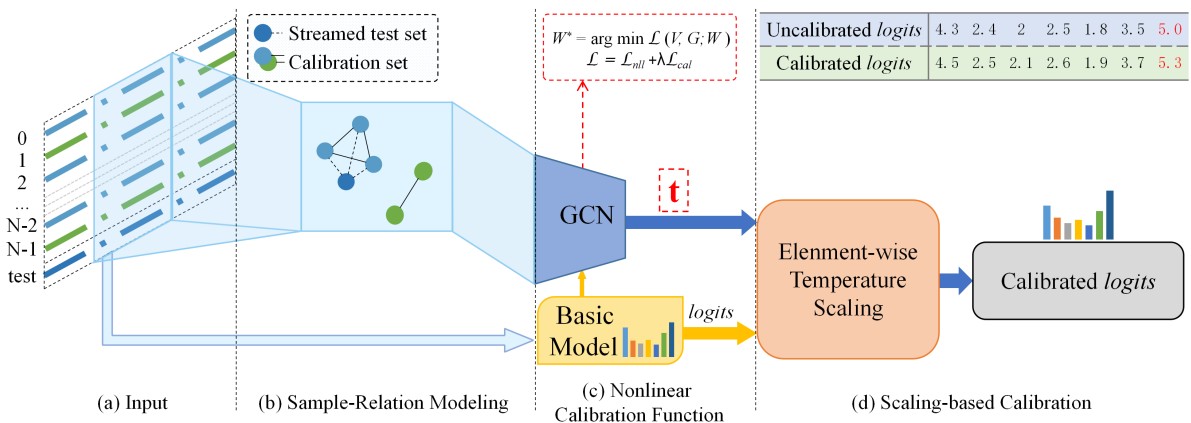

Figure 3: The overall framework of RECAL. The ranking of classes is unchanged after being calibrated.

enhance the model's ability to generate more reliable predictions for tabular datasets. As depicted in Figure 2, we now derive $G$ from a table, where the adjacency matrix is denoted by $\mathbf{A} \in \mathbb{R}^{N \times N}$.

### 3.2 Nonlinear Calibration Function

To explore the structured knowledge in $G$, we adopt Graph Convolutional Network (GCN) to propagate node features along the network topology and smooth the information between neighboring nodes through a nonlinear aggregation function (Giannakis et al., 2018). Firstly, the basic model $\mathcal{M}$ is adopted to compute logit for each sample in the held-out dataset before the softmax layer,

$$\mathbf{v_i} = \mathcal{M}(r_i), \tag{2}$$

where $i \in [1, N]$ and $\mathbf{V} = [\mathbf{v_1}, \ldots, \mathbf{v_N}]$ is obtained.

Then, we use Eq.3 to calculate the symmetric normalized Laplacian matrix:

$$\mathbf{L^{sys}} = \mathbf{D^{-1/2}LD^{-1/2}} = \mathbf{I} \text{ - } \mathbf{D^{-1/2}AD^{-1/2}}, \tag{3}$$

where $\mathbf{D} \in \mathbb{R}^{N \times N}$ is the degree matrix of graph $G$, and $\mathbf{I}$ is the unit matrix. For a $K$-classification task, the logit $\mathbf{v'_i}$ and the prediction probability $\mathbf{z_i}$ for sample $i$ after calibration can be calculated as:

$$\mathbf{V'} = \mathbf{L}^{sys}\mathbf{V}^{(l)}\mathbf{W}^{(l)} = [\mathbf{v'_1}, \ldots, \mathbf{v'_N}]^\top,$$
$$\mathbf{z_i} = [\sigma_{SM}(v'_{i,1}), \ldots, \sigma_{SM}(v'_{i,K})]^\top, \tag{4}$$

where $\mathbf{W}^{(l)}$ is the weight matrix of $l$-th layer in GCN, $\sigma_{SM}(v_{i,k}) = \frac{\exp(v_{i,k})}{\sum_{j=1}^{K} \exp(v_{i,j})}$ is the softmax operation, and the calibrated confidence is $\hat{p}_i = \max \mathbf{z_i}$, $k \in \{1, \ldots, K\}$. $v'_{i,k}$ can be any value so that $\hat{p}_i$ can traverse the interval $(\frac{1}{K}, 1)$, and the model can be calibrated correctly.

In summary, RECAL takes the logits $\mathbf{V}$ and the sample-relation graph $G$ as inputs to calibrate $\mathbf{V}$. During training, we use the validation set as the held-out calibration set $\mathcal{R}$. Then, $\mathbf{V}$ and $G$ are used to train the calibration model $\mathcal{C}$. During testing, we consider the real-time online scenario where the data to be calibrated comes in a streaming fashion with only a few samples at a time. To obtain more comprehensive graph structure information, we aggregate each incoming sample with the calibration set to construct a new graph structure $G_j^{test}, \forall j \in 1, \ldots, \left|\mathcal{R}^{test}\right|$.

### 3.3 Element-Wise Temperature Scaling

Now, we have a nonlinear calibration function to characterize the topological relations on the graph. However, the output of GCN directly as a calibrated logit cannot meet the requirement of accuracy maintenance. Temperature scaling is currently the most effective calibration method to maintain constant prediction accuracy. But, the temperature scaling performs the same linear transformation for each input data, which does not express the graph's topology. To this end, we propose element-wise temperature scaling to generate a unique temperature $t$ for each node in $G$.

As shown in Figure 3, RECAL first takes the output $\mathbf{V}$ and the sample-relation graph $G$ as input and outputs a temperature vector $\mathbf{t}$, where $t_i$ is unique for $\mathbf{v_i}$. Then, we use $t_i$ to transform the original logit $\mathbf{v_i}$ through scaling methods to obtain the calibrated logit $\mathbf{v'_i}$. Finally, the calibrated prediction result $\hat{p}_i$ is given by:

$$\mathbf{t} = \sigma^+(\mathbf{L}^{sys}\mathbf{V}^{(l)}\mathbf{W}^{(l)}) = [t_1, \ldots, t_N]^\top,$$
$$\mathbf{v'_i} = \mathbf{v}_i/t_i, \tag{5}$$
$$\hat{p}_i = \max \sigma_{SM}\left(\mathbf{v'_i}\right),$$

where $\sigma^+(t_i) = \log(1 + \exp(t_i))$ is known as element-wise softplus activation, which guarantees that $t_i \in \mathbb{R}$ is always greater than zero. When $t_i \to 0$, the predicted probability is infinitely close to 1. When $t_i \to \infty$, the probability $\hat{p}_i$ has the greatest uncertainty, and the predicted probability approaches $\frac{1}{K}$. Thus, RECAL can output any value on $(\frac{1}{K}, 1)$, and the basic model $\mathcal{M}$ can theoretically be perfectly calibrated.

RECAL is optimized with negative log-likelihood (NLL) (Hastie et al., 2009). This is a common indicator for measuring probabilistic models, minimizing NLL loss benefits uncertainty calibration. Given the one-hot label $\mathbf{y}_i = [y_{i,1}, \ldots, y_{i,K}]^\top$ and the prediction probability $\mathbf{z}_i$, the NLL loss over the calibration set can be calculated as:

$$\mathcal{L}_{nll} = -\sum_{i=1}^{N} \sum_{k=1}^{K} y_{i,k} \log(z_{i,k}). \qquad (6)$$

Inspired by Wang et al. (2021), we also add a regularization term

$$\mathcal{L}_{cal} = \frac{1}{n} \Big( \sum_{i=1}^{|cor|} 1 - z_{i,m}^{cor} + z_{i,s}^{cor} \\ + \sum_{i=1}^{|err|} z_{i,m}^{err} - z_{i,s}^{err} \Big), \qquad (7)$$

where $|cor|$ and $|err|$ are the number of samples correctly and incorrectly predicted and $z_{i,m}$ and $z_{i,s}$ are the max and submax prediction probability, to the loss function to explicitly reduce the confidence level of misclassified samples. The overall loss function is defined as:

$$\mathcal{L}_{\text{RECAL}} = \mathcal{L}_{nll} + \lambda \mathcal{L}_{cal}, \qquad (8)$$

where $\lambda$ is a hyperparameter to balance the effects of the regularization term.

# 4 Experiments

## 4.1 Datasets

To validate the effectiveness of RECAL, we conduct experiments on seven datasets with both implicit and explicit graph structures from the real-world —Bank Marketing (Moro et al., 2014), Qsar-Biodeg (Huang et al., 2020), Seismic-Bumps (Mansouri et al., 2013), County (Jia and Benson, 2020), House_Class (Pace and Barry, 1997), DBLP (Ren and Liu, 2020) and SLAP (Xiao et al., 2019). The

| Dataset | Datapoints | Features | N labels | Type |
|---|---|---|---|---|
| Bank Marketing | 45211 | 16 | 2 | Dense |
| Qsar-Biodeg | 1055 | 41 | 2 | Dense |
| Seismic-Bumps | 2583 | 18 | 2 | Dense |
| County | 3217 | 7 | 4 | Dense |
| House_Class | 20640 | 6 | 5 | Dense |
| DBLP | 14475 | 5002 | 4 | Sparse |
| SLAP | 20419 | 2701 | 15 | Sparse |

Table 1: Summary of datasets. It is worth noting that DBLP and SLAP datasets have an explicitly global graph structure, respectively. Yet in other datasets, we need to build graph by implicit relations.

data is split into training, validation, and testing sets in a ratio of 80%, 10%, and 10%, respectively. The details of the dataset are outlined in Table 1. To fully explain the graph construction process on each dataset, we have provided further instructions in Appendix A.2.

## 4.2 Experimental Settings

ECE (Naeini et al., 2015) and Brier Score as the evaluation metrics and the classical post-hoc method temperature scaling (TS) (Guo et al., 2017) as the baseline method. In addition, we compare a variant of our method, RECAL-MLP, which turns the GCN into a fully connected neural network. This is equivalent to a more complex matrix scaling (Guo et al., 2017). Appendix A.1 provides a more detailed description.

To demonstrate the generalizability of our method, we chose the classical machine learning method Logistic Regression (Wright, 1995), GBDT (Friedman, 2001), and a deep learning model called TabTransformer (Huang et al., 2020) as the basic model.

For Logistic Regression, we use scikit-learn's [1] official interface (Kramer, 2016) and modify the source code to ensure that the returned prediction distribution is the value before the softmax layer. We use the CatBoost (Prokhorenkova et al., 2018) as an implementation of GBDT. For each decision tree in CatBoost, we set its depth to 6, the early stopping rounds is 100, the number of epochs is 1000, $\|\lambda\|_2 = 0$. TabTransformer's implementation and parameter selection are consistent with (Huang et al., 2020). The implementation of temperature scaling is based on the original author's implementation (Guo et al., 2017). For our RECAL, we build a 2-layer GCN with the hidden layer of dimension 16. We set the learning rate as {0.001,

---
[1] https://scikit-learn.org/stable/

| Dataset | Calibration | Logistic | | | GBDT | | | Tab Transformer | | |
|---|---|---|---|---|---|---|---|---|---|---|
| | | $NLL(\downarrow)$ | $ECE(\downarrow)$ | $BS(\downarrow)$ | $NLL(\downarrow)$ | $ECE(\downarrow)$ | $BS(\downarrow)$ | $NLL(\downarrow)$ | $ECE(\downarrow)$ | $BS(\downarrow)$ |
| Bank Marketing | Uncal | $0.2603_{2.9}$ | $0.0248_{3.4}$ | $0.1539_{0.8}$ | $0.2082_{2.3}$ | $0.0159_{3.7}$ | $0.1281_{2.2}$ | $0.2119_{1.4}$ | $0.0141_{3.1}$ | $0.1391_{3.1}$ |
| | TS | $0.2602_{4.4}$ | $0.0235_{2.7}$ | $0.1539_{1.0}$ | $0.2071_{3.0}$ | $0.0158_{3.1}$ | $0.1274_{2.1}$ | $0.2105_{1.7}$ | $0.0090_{3.3}$ | $0.1387_{2.4}$ |
| | RECAL- MLP | $0.2591_{2.5}$ | $0.0208_{2.4}$ | $\mathbf{0.1532}_{0.9}$ | $0.2064_{4.1}$ | $0.0147_{4.3}$ | $0.1276_{4.1}$ | $0.2102_{2.3}$ | $0.0098_{4.2}$ | $\mathbf{0.1387}_{3.5}$ |
| | RECAL (ours) | $\mathbf{0.2537}_{2.1}$ | $\mathbf{0.0184}_{1.6}$ | $0.1549_{2.2}$ | $\mathbf{0.2063}_{3.9}$ | $\mathbf{0.0127}_{2.5}$ | $\mathbf{0.1272}_{4.0}$ | $\mathbf{0.2087}_{2.1}$ | $\mathbf{0.0075}_{2.3}$ | $0.1388_{3.6}$ |
| Qsar-Biodeg | Uncal | $0.2991_{2.4}$ | $0.0608_{4.1}$ | $0.1725_{3.5}$ | $0.3061_{2.6}$ | $0.0933_{4.9}$ | $0.1833_{3.3}$ | $0.2895_{5.2}$ | $0.0816_{6.7}$ | $0.1652_{1.9}$ |
| | TS | $0.2982_{0.7}$ | $0.0580_{3.3}$ | $0.1716_{2.8}$ | $0.2910_{3.5}$ | $0.0737_{5.0}$ | $0.1765_{3.9}$ | $0.2803_{4.1}$ | $0.0704_{3.8}$ | $0.1580_{2.3}$ |
| | RECAL- MLP | $0.2961_{1.5}$ | $0.0519_{4.7}$ | $0.1683_{2.0}$ | $0.2847_{3.2}$ | $0.0638_{6.3}$ | $0.1732_{3.7}$ | $0.2869_{4.0}$ | $0.0725_{2.2}$ | $0.1633_{1.5}$ |
| | RECAL (ours) | $\mathbf{0.2929}_{1.2}$ | $\mathbf{0.0504}_{3.6}$ | $\mathbf{0.1674}_{3.7}$ | $\mathbf{0.2816}_{3.0}$ | $\mathbf{0.0502}_{5.4}$ | $\mathbf{0.1717}_{2.8}$ | $\mathbf{0.2861}_{5.7}$ | $\mathbf{0.0600}_{7.6}$ | $\mathbf{0.1520}_{2.1}$ |
| Seismic-Bumps | Uncal | $0.1739_{2.8}$ | $0.0299_{4.2}$ | $0.0899_{0.9}$ | $0.1711_{3.3}$ | $0.0317_{5.0}$ | $0.0872_{1.4}$ | $0.1803_{4.0}$ | $0.0399_{4.8}$ | $0.0886_{1.6}$ |
| | TS | $0.1783_{1.3}$ | $0.0352_{2.6}$ | $0.0908_{0.2}$ | $0.1726_{3.8}$ | $0.0328_{4.1}$ | $0.0875_{1.0}$ | $0.1732_{4.1}$ | $0.0324_{5.9}$ | $0.0878_{2.1}$ |
| | RECAL- MLP | $0.1731_{0.3}$ | $0.0226_{3.5}$ | $\mathbf{0.0898}_{0.1}$ | $0.1675_{6.7}$ | $0.0283_{9.2}$ | $\mathbf{0.0868}_{2.2}$ | $0.1693_{3.9}$ | $0.0340_{4.6}$ | $0.0865_{1.3}$ |
| | RECAL (ours) | $\mathbf{0.1724}_{0.5}$ | $\mathbf{0.0216}_{3.4}$ | $0.0899_{0.2}$ | $\mathbf{0.1674}_{4.3}$ | $\mathbf{0.0243}_{1.1}$ | $0.0873_{0.1}$ | $\mathbf{0.1638}_{4.6}$ | $\mathbf{0.0212}_{5.6}$ | $\mathbf{0.0857}_{1.9}$ |
| County | Uncal | $1.2080_{2.9}$ | $0.0509_{9.2}$ | $0.6613_{2.7}$ | $0.9866_{2.3}$ | $0.0498_{5.1}$ | $0.5616_{3.0}$ | $1.1300_{5.1}$ | $0.0518_{5.1}$ | $0.6802_{2.4}$ |
| | TS | $1.2080_{1.1}$ | $0.0432_{5.3}$ | $0.6614_{0.7}$ | $0.9844_{2.0}$ | $0.0494_{4.7}$ | $0.5611_{3.4}$ | $1.1271_{5.8}$ | $0.0412_{3.8}$ | $0.6797_{2.7}$ |
| | RECAL- MLP | $1.2086_{0.4}$ | $0.0391_{6.8}$ | $0.6614_{1.5}$ | $0.9884_{3.3}$ | $0.0623_{6.2}$ | $0.5619_{2.7}$ | $1.1277_{4.1}$ | $0.0411_{4.9}$ | $0.6796_{1.8}$ |
| | RECAL (ours) | $\mathbf{1.2079}_{1.0}$ | $\mathbf{0.0385}_{7.8}$ | $\mathbf{0.6608}_{1.4}$ | $\mathbf{0.9828}_{2.8}$ | $\mathbf{0.0415}_{3.5}$ | $\mathbf{0.5604}_{1.9}$ | $\mathbf{1.1241}_{4.2}$ | $\mathbf{0.0389}_{3.6}$ | $\mathbf{0.6782}_{1.7}$ |
| House_Class | Uncal | $1.1916_{3.1}$ | $0.0178_{7.2}$ | $0.6008_{0.8}$ | $1.0983_{2.6}$ | $0.0366_{5.5}$ | $0.5690_{4.5}$ | $1.4577_{3.4}$ | $0.0220_{2.1}$ | $0.7392_{1.3}$ |
| | TS | $1.1936_{2.3}$ | $0.0275_{5.7}$ | $0.6051_{0.6}$ | $1.0904_{4.0}$ | $0.0178_{6.0}$ | $0.5664_{3.2}$ | $1.4560_{3.1}$ | $0.0189_{1.8}$ | $0.7390_{1.6}$ |
| | RECAL- MLP | $\mathbf{1.1829}_{4.2}$ | $0.0153_{6.2}$ | $0.6005_{1.0}$ | $1.0899_{1.9}$ | $0.0157_{4.1}$ | $0.5666_{3.8}$ | $1.4585_{2.9}$ | $0.0236_{2.3}$ | $0.7407_{1.4}$ |
| | RECAL (ours) | $1.1886_{8.1}$ | $\mathbf{0.0141}_{4.5}$ | $\mathbf{0.5964}_{1.2}$ | $\mathbf{1.0880}_{3.3}$ | $\mathbf{0.0154}_{3.8}$ | $\mathbf{0.5653}_{2.6}$ | $\mathbf{1.4540}_{2.0}$ | $\mathbf{0.0173}_{1.9}$ | $\mathbf{0.7371}_{1.3}$ |
| DBLP | Uncal | $0.6406_{2.1}$ | $0.0710_{4.2}$ | $0.2480_{1.2}$ | $0.5837_{5.7}$ | $0.1871_{8.3}$ | $0.2901_{1.7}$ | $1.1861_{3.8}$ | $0.1203_{1.8}$ | $0.6397_{3.1}$ |
| | TS | $0.6236_{1.7}$ | $0.0920_{5.5}$ | $0.2553_{1.6}$ | $0.4748_{4.2}$ | $0.0684_{4.3}$ | $0.2417_{1.1}$ | $1.1845_{3.7}$ | $0.0891_{5.0}$ | $0.6426_{3.9}$ |
| | RECAL- MLP | $\mathbf{0.5024}_{3.0}$ | $0.0714_{4.9}$ | $0.2445_{1.1}$ | $0.4628_{4.0}$ | $0.0405_{6.9}$ | $0.2340_{1.2}$ | $1.1770_{2.8}$ | $0.0973_{7.3}$ | $0.6394_{3.5}$ |
| | RECAL (ours) | $0.5089_{2.6}$ | $\mathbf{0.0584}_{4.5}$ | $\mathbf{0.2415}_{1.3}$ | $\mathbf{0.4569}_{3.9}$ | $\mathbf{0.0377}_{3.4}$ | $\mathbf{0.2301}_{1.6}$ | $\mathbf{1.1677}_{5.9}$ | $\mathbf{0.0718}_{2.6}$ | $\mathbf{0.6334}_{4.7}$ |
| SLAP | Uncal | $0.6992_{24}$ | $0.2757_{12}$ | $0.3314_{3.1}$ | $0.1535_{5.1}$ | $0.0817_{3.3}$ | $0.0606_{1.7}$ | $0.9016_{21}$ | $0.1933_{7.9}$ | $0.4798_{5.1}$ |
| | TS | $0.6454_{13}$ | $0.2144_{9.0}$ | $0.2655_{7.7}$ | $0.0873_{4.2}$ | $0.0234_{1.8}$ | $0.0501_{1.3}$ | $0.9014_{22}$ | $0.1926_{8.7}$ | $0.4785_{5.8}$ |
| | RECAL- MLP | $0.3429_{17}$ | $0.0896_{8.6}$ | $0.1611_{4.3}$ | $0.0688_{3.8}$ | $0.0083_{0.9}$ | $0.0450_{2.1}$ | $0.8630_{28}$ | $0.0932_{6.0}$ | $0.3922_{6.3}$ |
| | RECAL (ours) | $\mathbf{0.3231}_{19}$ | $\mathbf{0.0627}_{6.2}$ | $\mathbf{0.1598}_{4.8}$ | $\mathbf{0.0659}_{3.7}$ | $\mathbf{0.0061}_{0.3}$ | $\mathbf{0.0441}_{2.0}$ | $\mathbf{0.8581}_{36}$ | $\mathbf{0.0901}_{7.6}$ | $\mathbf{0.3765}_{4.8}$ |

Table 2: Summary of results using different calibration methods on different basic models and different datasets. Uncal indicates the basic models without calibration, BS represents the Brier Score, RECAL- MLP is the ablation study, and bold indicates the best result, the subscript of each result refers to the standard deviation $(\times 10^{-3})$.

0.01}, weight decay to 5e-3 for all datasets, $k$ is 5 in top-$k$. Our evaluation metrics are NLL loss, ECE, and Brier Score, all using official realizations (Naeini et al., 2015; Brier et al., 1950).

We use the validation set as the held-out calibration set. In the real world, models often need to predict consistently from streaming data. Therefore, we calibrate each row of data in the test set individually and construct a unique graph for each test sample by aggregating it with the validation set.

### 4.3  Main Results

The evaluation results of our model and all baselines are summarized in Table 2. The uncalibrated model usually has the highest values on all metrics, with ECE is typically between 1.4% to 20%, which indicates that the model used to predict the tabular data is poorly calibrated. This is not based on a specific basic model: we observed miscalibration on linear models, tree-based models, and transformer-based deep neural networks. In particular, the values of ECE are significantly worse on DBLP and SLAP, two sparse datasets with signifi-

cant graph structure.

It can be observed that TS performs well on TabTransformer which is due to the fact that the miscalibration of neural networks tends to be low-dimensional (Guo et al., 2017). But, it does not perform consistently enough on the traditional machine learning methods like GBDT and Logistic Regression (only 4 out of 7 datasets are successfully calibrated). This suggests that TS is unreliable for calibrating traditional machine learning methods. To explore the effect of the implicit sample relations between the data on the calibration results, we propose a variant RECAL-MLP of RECAL, in which the 2-layer GCN is replaced with a 2-layer MLP. On TabTransformer, TS performs better than RECAL-MLP. However, on the machine learning method, RECAL-MLP performs better than TS. This shows that applying an identical change to all logits on traditional machine learning methods is not enough.

Our method achieves the best results on all datasets. In particular, compared to the temperature scaling, our method reduces the ECE by 20%, 32%, 26%, 13%, 17%, 45%, and 74% on the

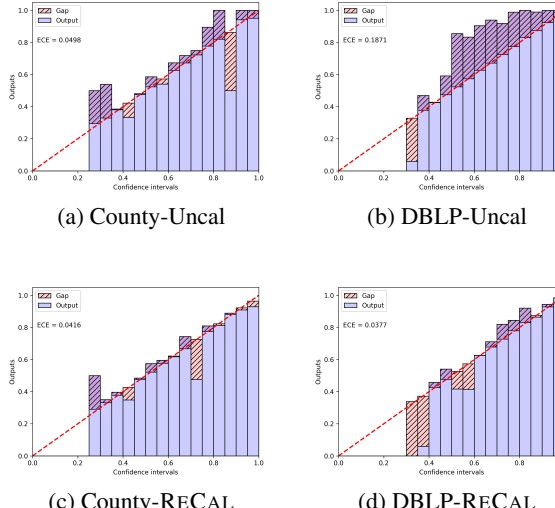

(a) County-Uncal      (b) DBLP-Uncal

(c) County-ReCal      (d) DBLP-ReCal

Figure 4: Reliability diagrams for GBDT before and after confidence calibration on County and DBLP datasets. Any deviation from a perfect(i.e., the gap) diagonal represents miscalibration.

Bank, Qsar-Biodeg, Seismic-Bump, House_Class, County, DBLP, SLAP datasets, respectively, on the GBDT model. On TabTransformer, compared to the temperature scaling, our method reduces the ECE by 17%, 15%, 35%, 8%, 6%, 19%, and 45%, respectively. On Logistic Regression, compared to the temperature scaling, our method reduces the ECE by 22%, 17%, 39%, 49%, 11%, 37%, and 71%, respectively. NLL and Brier Score show the same trend. In brief, our method outperforms the baseline method and the graphs provide valuable information for confidence calibration.

### 4.4 Reliability Diagrams

We further depict the reliability diagrams (Guo et al., 2017) to evaluate the quality of uncertainty estimation. As shown in Figure 4, the confidence range is equally divided into 20 bins. Then we calculate the average accuracy of each bin. The red diagonal represents the perfectly calibrated line, and its confidence indicates the probability of the model predicting correctly. For example, the mean value of the diagonal function in the bin $[0.95, 1.0]$ is 0.97, and then the classification accuracy of the nodes in this bin should be 97%.

We can see that the average precision of most boxes of the uncalibrated model is higher than the diagonal, indicating that GBDT always outputs a low confidence, i.e., GBDT are usually underconfident. Compared with uncalibrated GBDT, RE-

| Dataset | Calibraction | Noise Rate % | | | | |
|---|---|---|---|---|---|---|
| | | 0 | 10 | 20 | 30 | 50 |
| Bank Marketing | TS | 0.0158 | 0.0248 | 0.0347 | 0.0467 | 0.0672 |
| | ReCal | **0.0127** | **0.0212** | **0.0315** | **0.0457** | **0.0646** |
| County | TS | 0.0499 | 0.0524 | 0.0592 | 0.0551 | 0.0815 |
| | ReCal | **0.0415** | **0.0476** | **0.0464** | **0.0525** | **0.0677** |
| House_Class | TS | 0.0178 | 0.0311 | 0.0623 | 0.0845 | 0.1326 |
| | ReCal | **0.0154** | **0.0296** | **0.0566** | **0.0817** | **0.1265** |

Table 3: ECE of ReCal and TS with noisy data on GBDT.

| Dataset | TabTransformer | | |
|---|---|---|---|
| | Uncal | TS | ReCal |
| Bank Marketing | 0.0397 | 0.0488 | **0.0317** |
| Seismic-Bump | 0.0352 | 0.0525 | **0.0298** |

Table 4: ECE of ReCal and TS with distribution shift on TabTransformer.

CAL most closely recovers the desired diagonal function, and each of the bins are well calibrated. This suggests that ReCal can yield better confidence estimates, showing it is more trustworthy.

### 4.5 The Robustness of ReCal

We further explored the robustness of ReCal to noisy data and distribution shift against the baseline TS.

**Noisy Data.** To obtain the noisy data, we first select the coninuous feature that has the largest impact on the tree-base model among all features. In CatBoost, the impact factor of each feature can be obtained from *model.feature_importances_*. Next, on the test set, we contaminate the data by replacing a number of values with random ones in the selected columns. We input the noisy streamed data into the trained GBDT model to obtain the output and calibrate it using the trained ReCal model. Table 3 gives the results of the calculations on three different datasets. As the noisy rate increases, ReCal performs better in ECE and thus is more robust than temperature scaling. We conjecture that the graph brings robustness because its topology dilutes the effect of noise to a certain extent.

**Distribution Shift.** Distribution shift refers to the change in the underlying data distribution between training and testing phases in machine learning, which can affect model performance when the model encounters data that differs significantly from what it has learned during training. We first partition the dataset into two parts with different

| Dataset | Logistic | | GDBT | | TabTransformer | |
|---|---|---|---|---|---|---|
| | online | offline | online | offline | online | offline |
| Bank Marketing | 0.0184 | **0.0178** | 0.0127 | **0.0106** | **0.0075** | 0.0081 |
| Qsar-Biodeg | 0.0504 | **0.0447** | **0.0502** | 0.0511 | 0.0600 | **0.0479** |
| Seismic-Bumps | 0.0216 | **0.0175** | 0.0243 | **0.0195** | 0.0212 | **0.0203** |
| County | 0.0385 | **0.0361** | 0.0415 | **0.0385** | **0.0389** | 0.0413 |
| House_Class | **0.0141** | 0.0148 | 0.0154 | **0.0147** | 0.0173 | **0.0135** |
| DBLP | **0.0584** | 0.0608 | 0.0377 | **0.0295** | **0.0718** | 0.0835 |
| SLAP | 0.0627 | **0.0626** | 0.0061 | **0.0051** | 0.0901 | **0.0890** |

Table 5: ECE of RECAL in online and offline scenarios.

| Dataset | Logistic | | |
|---|---|---|---|
| | Uncal | TS | RECAL |
| Qsar-Biodeg | 136.5013 | 129.5814 | **119.8235** |
| Bank Marketing | 6802.1870 | 6750.9567 | **6548.1597** |
| County | 57.3955 | 56.5735 | **53.5246** |
| SLAP | 7.7824 | 5.0452 | **0.0653** |

Table 6: Total variation of confidence before and after calibration for four datasets on Logistic Regression. Uncal indicates uncalibrated, TS indicates temperature scaling and bold indicates the best result.

distributions based on an artificially chosen feature. We conduct experiments on Bank and Seismic-Bumps. For Bank, we divide it into two subsets with and without houses based on *housing*. For Seismic-Bumps, we partition the dataset according to *seismic*, which grades the hazard of the detected seismic waves. We make the training and validation sets have the same distribution, and the test set obeys a different distribution. Then, we train the basic model using the training set, train RECAL utilizing the validation set, and test it using the test set. Results on the two datasets are presented in Table 4. We can see the same conclusion as in the case of noisy data, i.e. that RECAL shows excellent stability and can still calibrate the model, while TS cannot handle this case.

### 4.6 Offline Test Scenario

Further, we conduct an experiment to show that RECAL is equally effective in the offline scenario. In this scenario, uncalibrated samples arrive at the same time. We also use the validation set to train the RECAL. Unlike the online scenario that generates a unique graph for each test sample, we use all the test samples to construct one graph and get the calibration results for the whole test set by forward propagation once. As shown in Table 5, RECAL works equally well in the offline scenario and performs better than online in most cases. We believe this is because building the graph on the test set allows for a more accurate characterization of its sample relation.

### 4.7 Why RECAL Achieves Well Calibration?

To validate the effectiveness of the graph guided by implicit sample relations in enhancing confidence calibration, we perform the following experiment. We utilize temperature scaling as the calibration function and calculate the total change in confidence, which represents the cumulative difference in confidence values between adjacent nodes within

the graph $G$. Table 6 presents the results comparing the total confidence change before and after calibration. Our research findings demonstrate that applying temperature scaling decreases the total change in confidence. This observation supports the assertion that well-calibrated basic models result in increased similarity in confidence levels between nodes with the defined relation after calibration. GNN performs massage passing along its topology, making it smoother between neighboring nodes.Thus, our approach of incorporating implicit sample relations into the graph construction facilitates more reliable predictions and improves confidence calibration for tabular datasets.

## 5   Conclusion

In this paper, we exploit the implicit sample relation on tabular data and propose a post-hoc calibration framework RECAL, which improves the confidence of the model while maintaining classification accuracy. It is worth noting that our approach is general and applicable to both deep learning models and traditional machine learning models. RECAL uses the GCN as a nonlinear calibration function so that the confidence can be propagated along the graph between the sample. In addition, RECAL generates a unique temperature for each sample to be calibrated using the scaling method, which has the accuracy-preserving property. We perform detailed experiments on binary and multi-classified tabular datasets with implicit and explicit graph. Empirically results demonstrate that RECAL has a better performance compared to traditional calibration models. In addition, experiments demonstrate that RECAL has remarkable robustness against data noise and distribution shift. An interesting future research work would be to use embedding of samples to guide the discovery of graph $G$. This extension would enable the application of RECAL not only to tabular data but also to textual and image data.

## Limitations

Our work is the first attempt to use graph structures to represent the relation between samples in tabular data for improving the reliability of tabular model. Using sample-relation graph, RECAL outputs a unique temperature $t$ for each sample, and then finely calibrates each sample using a scaling-based method to make the model more trustworthy. Further work can apply the idea of modeling implicit sample relations from datasets to enhance the task to more domains.

In our analysis, we select a specific feature by a GBDT model pre-trained on a calibration set as the basis for our definition of implicit sample relations. However, in real scenarios, large tables with hundreds or thousands of features may exist. Meanwhile, the method of traversing each node to connect neighboring nodes makes the overhead of building the graph large. In future research, this can be done by expressing each sample as an embedding vector and then using a clustering algorithm (MacQueen, 1967; Ashraf et al., 2021) to build a graph on the dataset. With the use of embedding, the method is also no longer limited to tabular data, which opens up new ideas for improving the trustworthiness of machine learning.

## Acknowledgements

We sincerely thank all the anonymous reviewers for providing valuable feedback. This work is supported by the National Key Research and Development Program of China (Grant No. 2021YFB0300104), the youth program of National Science Fund of Tianjin, China (Grant No. 22JCQNJC01340), and the Fundamental Research Funds for the Central University, Nankai University (Grant No. 63221028)

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

# A Appendix

## A.1 Evaluation Metrics and Baseline

**Evaluation Metrics.** The expected calibration error, also called ECE divides the prediction into B equally spaced bins (Naeini et al., 2015). It calculates the weighted average of the accuracy\confidence difference of the samples in each bin. We use ECE as the most important evaluation indicator. The calculation equation is as follows:

$$\mathbf{ECE} = \sum_{i=1}^{B} \frac{|B_i|}{|\mathcal{R}^{test}|} \left| \text{acc}(B_i) - \text{conf}(B_i) \right|, \quad (9)$$

where $|\mathcal{R}^{test}|$ is the total number of test sets, $B_i$ represents the $i$-th bin, $|B_i|$ represents the number of samples in the i-th bin. $\text{acc}(B_i)$ denotes the accuracy, $\text{conf}(B_i)$ represents the average of confidences in the $i$-th bin. The Brier Score (Brier et al., 1950) is also a measure of the degree of calibration. It portrays the difference in predicted probability from the true label. More precisely,

$$\mathbf{BS} = \frac{1}{|\mathcal{R}^{test}|} \sum_{n=1}^{|\mathcal{R}^{test}|} (\hat{p}_n - y_n)^2, \quad (10)$$

where $y_n$ is the label of the $n$-th sample.

**Baseline.** As a comparison, we select the classical post-hoc method temperature scaling (TS) (Guo et al., 2017) as our baseline, which is given by

$$\hat{p}_i = \max \sigma_{SM}(\mathbf{v_i}/t). \quad (11)$$

## A.2 Dataset Details

Here, we provide a detailed description of the dataset used in this paper, as well as the basis for creating the graphs.

- **Bank Marketing** (Moro et al., 2014) is related to the direct marketing activities of a Portuguese banking institution, which contains 45211 samples, and the classification goal is to predict if the client will subscribe (yes/no) a term deposit. Guided by the pre-trained GBDT, we select *month* as the basis for building the graph. Since it is a categorical feature, we connect samples with the same value.

- **Seismic-Bumps** (Huang et al., 2020) predict whether a high energy seismic bump will occur in the next shift by using the detection data

in the mine, which contains 2583 samples. It is a binary dataset. Based on the contribution to the classification results, we use *gpuls*, the number of pulses recorded by the most active geophones in the previous transformation, to build a graph with each node connecting the five closest nodes according to the value of *gpuls*.

- **Qsar-Biodeg** (Mansouri et al., 2013) dataset counts the chemical structure of molecules to predict whether they are readily biodegradable (RB/NRB), which contains 1058 samples. By pre-training a GBDT model using the validation set, we use *SpMax_B* as the basis for building the graph.

- **House_Class** (Pace and Barry, 1997) is collected from the 1990 census, which contains 20640 samples. The original dataset is a regression dataset with the predicted outcome being the price of the property. To construct a multi-classification tabular dataset, we divide the prediction results into five intervals $[1, 1.5, 2, 2.5]$ (Ivanov and Prokhorenkova, 2021). For each block, connect the five closest blocks within a given distance, measured by latitude and longitude.

- **County** (Jia and Benson, 2020) is a county-level map dataset containing 3217 samples. We divide the unemployment rate into four boxes as prediction labels, <0.4, 0.4-0.5, 0.5-0.7, >0.7. Each node is a county, and if two nodes border each other, they are connected.

- **SLAP** (Xiao et al., 2019) is a sparse multi-node-type dataset in the domain of bioinformatics with node types including genes, diseases, chemical compounds, etc. Our goal is to predict one of the 15 gene types. It contains 20419 samples. We use the same graph structure as Ivanov and Prokhorenkova (2021).

- **DBLP** (Ren and Liu, 2020) is a sparse classification dataset with 14475 samples. It predicts the author's research direction and is divided into four groups(database, information retrieval, data mining, and machine learning). We use the same graph structure as Ivanov and Prokhorenkova (2021).