# OpenReview forum: "RECAL: Sample-Relation Guided Confidence Calibration over Tabular Data"
_EMNLP/2023/Conference — EMNLP 2023 Findings_

### Official Review · Reviewer_tBxh · 2023-08-05

**Soundness:** 4

**Excitement:**

3: Ambivalent: It has merits (e.g., it reports state-of-the-art results, the idea is nice), but there are key weaknesses (e.g., it describes incremental work), and it can significantly benefit from another round of revision. However, I won't object to accepting it if my co-reviewers champion it.

**Paper Topic And Main Contributions:**

The paper proposes to calibrate the probabilities of a classifier that operates over tabular data by incorporating the structure within the table learnt using gradient boosted decision tree and a graph convolutional network on a validation set.


**Reasons To Accept:**

Outperforms temperature scaling and MLP -based post-processed calibration efforts on 7 tasks, across 3 base models and metrics

**Reasons To Reject:**

Utilizing the GBDT features in a Platt-scaling equivalent using a linear regression recalibration is missing

**Reproducibility:**

4: Could mostly reproduce the results, but there may be some variation because of sample variance or minor variations in their interpretation of the protocol or method.

**Reviewer Confidence:**

3: Pretty sure, but there's a chance I missed something. Although I have a good feel for this area in general, I did not carefully check the paper's details, e.g., the math, experimental design, or novelty.

**Typos Grammar Style And Presentation Improvements:**

Abstract: s/wildly/widely; lack of/lack,

many other typos throughout the text

---

> ### Author Rebuttal · Authors · 2023-08-28
>
> __Q1:__ Utilizing the GBDT features in a Platt-scaling equivalent using a linear regression recalibration is missing
>
> __A1:__ Thanks for pointing this out. Following the suggestion of reviewer 1, we use Isotonic Regression as a calibration function to compare with RECAL on three datasets: Bank Marketing, Qsar-Biodeg and Seismic-Bumps. The results of the experiments on three basic models are shown in the table below, where  Uncal indicates the basic models without calibration, BS represents the Brier Score, IR represents the Isotonic Regression, and bold indicates the best result, the subscript of each result refers to the standard deviation ( ×10$^{−3}$ ).
>
> **Logistic:**
> | Dataset           | Calibration         |    NLL(↓)       |    ECE(↓)    |   BS(↓)   |
> |-------------------|---------------------|-----------------|---------------|----------------|
> |        | Uncal        |    0.2603$_{2.9}$    |    0.0248$_{3.4}$   |   0.1539$_{0.8}$ |
> |    Bank Marketing    | IR        |    0.2545$_{1.1}$    |    0.02069$_{0.8}$   |   **0.1491$_{0.4}$** |
> |        | RECAL        |   **0.2537$_{2.1}$**    |    **0.0184$_{1.6}$**   |   0.1549$_{2.2}$  |
> |               |           |           |          |               |
> |               |    Uncal       |     0.2991$_{2.4}$      |     0.0608$_{4.1}$    |        0.1725$_{3.5}$        |
> |       Qsar-Biodeg        |      IR     |      0.2937$_{2.0}$     |      0.0604$_{3.2}$     |       0.1734$_{3.1}$        |
> |               |       RECAL    |     **0.2929$_{1.2}$**     |    **0.0504$_{3.6}$**     |        **0.1674$_{3.7}$**      |
> |               |           |           |          |               |
> |               |     Uncal      |      0.1739$_{2.8}$     |    0.0299$_{4.2}$     |     0.0899$_{0.9}$           |
> |         Seismic-Bumps      |    IR       |     0.1728$_{2.6}$       |     0.0290$_{1.7}$      |        **0.0882$_{0.4}$**       |
> |               |      RECAL     |     **0.1724$_{0.5}$**       |    **0.0216$_{3.4}$**     |     0.0899$_{0.2}$          |
>
> **GBDT:**
>
> | Dataset           | Calibration         |    NLL(↓)       |    ECE(↓)    |   BS(↓)   |
> |-------------------|---------------------|-----------------|---------------|----------------|
> |        | Uncal        |    0.2082$_{2.3}$    |    0.0159$_{3.7}$   |   0.1281$_{2.2}$ |
> |    Bank Marketing    | IR        |    0.2031$_{1.3 }$    |    0.0138$_{0.7}$   |   0.1279$_{2.1}$ |
> |        | RECAL        |   **0.2063$_{3.9}$**    |    **0.0127$_{2.5}$**   |   **0.1272$_{4.0}$**  |
> |               |           |           |          |               |
> |               |    Uncal       |     0.3061$_{2.6}$      |     0.0933$_{4.9}$    |        0.1833$_{3.3}$        |
> |       Qsar-Biodeg        |     IR     |   0.2993$_{1.9}$     |      0.0647$_{2.9}$     |       0.1725$_{7.1}$        |
> |               |       RECAL    |     **0.2816$_{3.0}$**     |    **0.0502$_{5.4}$**     |      **0.1717$_{2.8}$**      |
> |               |           |           |          |               |
> |               |     Uncal      |      0.1711$_{3.3}$     |    0.0317$_{5.0}$     |     **0.0872$_{1.4}$**            |
> |         Seismic-Bumps      |    IR       |     0.1683$_{5.4}$       |     0.0277$_{2.6}$      |        0.0881$_{3.2}$      |
> |               |      RECAL     |     **0.1674$_{4.3}$**       |    **0.0243$_{1.1}$**     |     0.0873$_{0.1}$          |
>
> **Tab Transformer:**
>
> | Dataset           | Calibration         |    NLL(↓)       |    ECE(↓)    |   BS(↓)   |
> |-------------------|---------------------|-----------------|---------------|----------------|
> |        | Uncal        |    0.2119$_{1.4}$    |    0.0141$_{3.1}$   |   0.1391$_{3.1}$ |
> |    Bank Marketing    | IR        |    0.2117$_{2.5}$    |    0.0111$_{1.8}$   |   0.1390$_{2.5}$ |
> |        | RECAL        |   **0.2087$_{2.1}$**    |    **0.0075$_{2.3}$**   |   **0.1388$_{3.6}$**  |
> |               |           |           |          |               |
> |               |    Uncal       |     0.2895$_{5.2}$      |     0.0816$_{6.7}$    |        0.1652$_{1.9}$        |
> |       Qsar-Biodeg        |     IR     |   0.2845$_{3.5}$     |      0.0796$_{2.6}$     |       0.1602$_{0.7}$        |
> |               |       RECAL    |     **0.2861$_{5.7}$**     |    **0.0600$_{7.6}$**     |      **0.1520$_{2.1}$**      |
> |               |           |           |          |               |
> |               |     Uncal      |      0.1803$_{4.0}$     |    0.0399$_{4.8}$     |     0.0886$_{1.6}$           |
> |         Seismic-Bumps      |    IR       |     0.1642$_{5.2}$       |     0.0261$_{2.3}$      |        0.0866$_{1.5}$      |
> |               |      RECAL     |     **0.1638$_{4.6}$**       |    **0.0212$_{5.6}$**     |     **0.0857$_{1.9}$**          |
>
> As can be seen from the results, RECAL outperforms IR in most cases. Additionally, Histogram Binning changes the ranking of the model's predictive probabilities, which is inconsistent with our goal of not changing the accuracy of the model itself. Platt Scaling is often used to calibrate binary classification tasks, and our baseline TS has been shown to be superior to Platt Scaling (Guo et al., 2017). We will also keep an eye on the latest developments in scaling-based calibration methods and compare them with RECAL on tabular data.
>
> References:
>
> Chuan Guo, Geoff Pleiss, Yu Sun, and Kilian Q Weinberger. 2017. On calibration of modern neural networks. In International conference on machine learning, pages 1321–1330. PMLR.
>
> __Q2:__ Typos Grammar Style And Presentation Improvements
>
> __A2:__ We appreciate your kind suggestions. We will revise the manuscript thoroughly and carefully.

---

### Official Review · Reviewer_nFCr · 2023-08-11

**Typos Grammar Style And Presentation Improvements:** none
**Soundness:** 4

**Excitement:**

3: Ambivalent: It has merits (e.g., it reports state-of-the-art results, the idea is nice), but there are key weaknesses (e.g., it describes incremental work), and it can significantly benefit from another round of revision. However, I won't object to accepting it if my co-reviewers champion it.

**Missing References:**

none

**Paper Topic And Main Contributions:**

This paper targets at the confidence estimation of machine learning models applied to tabular data, which contains implicit sample relations. The proposed method introduces a general post-training confidence calibration framework by employing graph neural networks. This model employs existing GBDT to select the most significant feature and then classify given samples with the feature for graph construction. Finally, this method adopt GCN as a nonlinear calibration function to explore the graph information and obtain normalized scores. Different from previous methods, element-wise temperature scaling is proposed to calculate the calibrated prediction result. Extensive experiments on tabular datasets show the superiority of the proposed method against another method(just a method).

**Questions For The Authors:**

Is there another better approach here regarding this area? I think it seems like the idea is fairly easy to understand and think about but really novel, so there really hasn't been any work on it before?

**Reasons To Accept:**

1. The paper is well written especially introduction and related work and makes it easy to understand.
2. I should mention right off the bat that I don't have much experience in this specific field. But I think the prospect of considering the study of relationships between different samples of data from the table is very promising. So if as the authors say they are the first to propose the method, I think it is very novel. It also depends on the opinion of other fellow reviewers.
3. This article conducts experiments on a large number of datasets to illustrate the validity of its methods. Analysis for confidence estimation of result is complete.


**Reasons To Reject:**

1.	Only two methods were compared in the experiment, There was no comparison with the most recently published method, which may have overstated the promise of this approach
2. The authors do not provide some source code to demo the algorithm so reproducibility is questionable.



**Reproducibility:**

4: Could mostly reproduce the results, but there may be some variation because of sample variance or minor variations in their interpretation of the protocol or method.

**Reviewer Confidence:**

2: Willing to defend my evaluation, but it is fairly likely that I missed some details, didn't understand some central points, or can't be sure about the novelty of the work.

---

> ### Author Rebuttal · Authors · 2023-08-28
>
> __Q1:__ Only two methods were compared in the experiment, There was no comparison with the most recently published method, which may have overstated the promise of this approach
>
> __A1:__ Thanks for pointing this out. Following the suggestion of reviewer 1, we use Isotonic Regression as a calibration function to compare with RECAL on the three datasets: Bank Marketing, Qsar-Biodeg and Seismic-Bumps. The results of the experiments on three basic models are shown in the table below, where Uncal indicates the basic models without calibration, BS represents the Brier Score, IR represents the Isotonic Regression, and bold indicates the best result, the subscript of each result refers to the standard deviation ( ×10−3 ).
>
> **Logistic:**
> | Dataset           | Calibration         |    NLL(↓)       |    ECE(↓)    |   BS(↓)   |
> |-------------------|---------------------|-----------------|---------------|----------------|
> |        | Uncal        |    0.2603$_{2.9}$    |    0.0248$_{3.4}$   |   0.1539$_{0.8}$ |
> |    Bank Marketing    | IR        |    0.2545$_{1.1}$    |    0.02069$_{0.8}$   |   **0.1491$_{0.4}$** |
> |        | RECAL        |   **0.2537$_{2.1}$**    |    **0.0184$_{1.6}$**   |   0.1549$_{2.2}$  |
> |               |           |           |          |               |
> |               |    Uncal       |     0.2991$_{2.4}$      |     0.0608$_{4.1}$    |        0.1725$_{3.5}$        |
> |       Qsar-Biodeg        |      IR     |      0.2937$_{2.0}$     |      0.0604$_{3.2}$     |       0.1734$_{3.1}$        |
> |               |       RECAL    |     **0.2929$_{1.2}$**     |    **0.0504$_{3.6}$**     |        **0.1674$_{3.7}$**      |
> |               |           |           |          |               |
> |               |     Uncal      |      0.1739$_{2.8}$     |    0.0299$_{4.2}$     |     0.0899$_{0.9}$           |
> |         Seismic-Bumps      |    IR       |     0.1728$_{2.6}$       |     0.0290$_{1.7}$      |        **0.0882$_{0.4}$**       |
> |               |      RECAL     |     **0.1724$_{0.5}$**       |    **0.0216$_{3.4}$**     |     0.0899$_{0.2}$          |
>
> **GBDT:**
>
> | Dataset           | Calibration         |    NLL(↓)       |    ECE(↓)    |   BS(↓)   |
> |-------------------|---------------------|-----------------|---------------|----------------|
> |        | Uncal        |    0.2082$_{2.3}$    |    0.0159$_{3.7}$   |   0.1281$_{2.2}$ |
> |    Bank Marketing    | IR        |    0.2031$_{1.3 }$    |    0.0138$_{0.7}$   |   0.1279$_{2.1}$ |
> |        | RECAL        |   **0.2063$_{3.9}$**    |    **0.0127$_{2.5}$**   |   **0.1272$_{4.0}$**  |
> |               |           |           |          |               |
> |               |    Uncal       |     0.3061$_{2.6}$      |     0.0933$_{4.9}$    |        0.1833$_{3.3}$        |
> |       Qsar-Biodeg        |     IR     |   0.2993$_{1.9}$     |      0.0647$_{2.9}$     |       0.1725$_{7.1}$        |
> |               |       RECAL    |     **0.2816$_{3.0}$**     |    **0.0502$_{5.4}$**     |      **0.1717$_{2.8}$**      |
> |               |           |           |          |               |
> |               |     Uncal      |      0.1711$_{3.3}$     |    0.0317$_{5.0}$     |     **0.0872$_{1.4}$**            |
> |         Seismic-Bumps      |    IR       |     0.1683$_{5.4}$       |     0.0277$_{2.6}$      |        0.0881$_{3.2}$      |
> |               |      RECAL     |     **0.1674$_{4.3}$**       |    **0.0243$_{1.1}$**     |     0.0873$_{0.1}$          |
>
> **Tab Transformer:**
>
> | Dataset           | Calibration         |    NLL(↓)       |    ECE(↓)    |   BS(↓)   |
> |-------------------|---------------------|-----------------|---------------|----------------|
> |        | Uncal        |    0.2119$_{1.4}$    |    0.0141$_{3.1}$   |   0.1391$_{3.1}$ |
> |    Bank Marketing    | IR        |    0.2117$_{2.5}$    |    0.0111$_{1.8}$   |   0.1390$_{2.5}$ |
> |        | RECAL        |   **0.2087$_{2.1}$**    |    **0.0075$_{2.3}$**   |   **0.1388$_{3.6}$**  |
> |               |           |           |          |               |
> |               |    Uncal       |     0.2895$_{5.2}$      |     0.0816$_{6.7}$    |        0.1652$_{1.9}$        |
> |       Qsar-Biodeg        |     IR     |   0.2845$_{3.5}$     |      0.0796$_{2.6}$     |       0.1602$_{0.7}$        |
> |               |       RECAL    |     **0.2861$_{5.7}$**     |    **0.0600$_{7.6}$**     |      **0.1520$_{2.1}$**      |
> |               |           |           |          |               |
> |               |     Uncal      |      0.1803$_{4.0}$     |    0.0399$_{4.8}$     |     0.0886$_{1.6}$           |
> |         Seismic-Bumps      |    IR       |     0.1642$_{5.2}$       |     0.0261$_{2.3}$      |        0.0866$_{1.5}$      |
> |               |      RECAL     |     **0.1638$_{4.6}$**       |    **0.0212$_{5.6}$**     |     **0.0857$_{1.9}$**          |
>
> As can be seen from the results, RECAL outperforms IR in most cases. Additionally, Histogram Binning changes the ranking of the model's predictive probabilities, which is inconsistent with our goal of not changing the accuracy of the model itself. Also, we investigated recent work and to the best of our knowledge, we are the first to use implicit relationships between tabular data to guide model calibration with a scaling-based approach. We found that traditional machine learning methods applied to tabular datasets are also in need of calibration. Unlike current calibration methods that focus only on deep neural networks, our approach is equally valid on traditional machine learning methods.
>
> __Q2:__ The authors do not provide some source code to demo the algorithm so reproducibility is questionable.
>
> __A2:__ Due to time limitation, the code was not organized previously. We will clean up the code in the git repository and make it public after the paper is accepted. In addition, we call the LogisticRegression in sklearn, in order to get the logits before softmax, users need to modify the source code in the environment. To avoid user changing the environment, we are replacing the previous calls to sklearn with our manual implementation of LogisticRegression, which will also be made available in the git repository after the paper is accepted.
>
> __Q3:__ Is there another better approach here regarding this area? I think it seems like the idea is fairly easy to understand and think about but really novel, so there really hasn't been any work on it before?
>
> __A3:__ Thank you for your question. After our careful survey, we believe that we are indeed the first work to focus on model calibration applied to tabular data. Unlike previous work, we are the first to attempt to use implicit relationships between tabular data to guide model calibration, which is generalizable to a wide range of models applied on tabular data. We believe that our work is important for improving the fairness and reliability of machine learning on tabular datasets.

---

### Official Review · Reviewer_XNjM · 2023-08-19

**Soundness:** 4

**Excitement:**

4: Strong: This paper deepens the understanding of some phenomenon or lowers the barriers to an existing research direction.

**Paper Topic And Main Contributions:**

This paper discusses the challenge in current machine learning models for tabular data where they lack accurate confidence estimation, crucial for high-risk applications like financial fraud detection. The study reveals that real-world tabular datasets inherently have implicit sample relations that can aid in more precise estimation. To address this, the authors introduce RECAL, a post-training confidence calibration framework that employs graph neural networks. This framework captures the relationships between different samples, improving calibration quality when compared to methods that disregard these sample relations.

**Reasons To Accept:**

* The paper's central idea of utilizing graph neural networks to comprehend data relationships for aiding calibration is really interesting.
* The way the paper is presented and explained is clear and easy to follow.
* It's great to see better results compared to the widely used temperature scaling technique.

**Reasons To Reject:**

A minor concern, but it would have been beneficial to see comparisons with other calibration mechanisms such as isotonic regression, histogram binning, and platt scaling.

**Reproducibility:**

3: Could reproduce the results with some difficulty. The settings of parameters are underspecified or subjectively determined; the training/evaluation data are not widely available.

**Reviewer Confidence:**

4: Quite sure. I tried to check the important points carefully. It's unlikely, though conceivable, that I missed something that should affect my ratings.

---

> ### Author Rebuttal · Authors · 2023-08-28
>
> __Q1:__ A minor concern, but it would have been beneficial to see comparisons with other calibration mechanisms such as isotonic regression, histogram binning, and platt scaling.
>
> __A1:__ We appreciate your kind suggestions. We supplement the experiments using Isotonic Regression（IR） for calibration on the Bank Marketing, Qsar-Biodeg and Seismic-Bumps datasets. The results of the experiments on three basic models are shown in the table below, where  Uncal indicates the basic models without calibration, BS represents the Brier Score, IR represents the Isotonic Regression, and bold indicates the best result, the subscript of each result refers to the standard deviation ( ×10$^{−3}$ ). As can be seen from the results, RECAL outperforms IR in most cases. We will conduct more extensive experimental analyses and refine the paper during the Camera-ready phase.
>
> **Logistic:**
> | Dataset           | Calibration         |    NLL(↓)       |    ECE(↓)    |   BS(↓)   |
> |-------------------|---------------------|-----------------|---------------|----------------|
> |        | Uncal        |    0.2603$_{2.9}$    |    0.0248$_{3.4}$   |   0.1539$_{0.8}$ |
> |    Bank Marketing    | IR        |    0.2545$_{1.1}$    |    0.02069$_{0.8}$   |   **0.1491$_{0.4}$** |
> |        | RECAL        |   **0.2537$_{2.1}$**    |    **0.0184$_{1.6}$**   |   0.1549$_{2.2}$  |
> |               |           |           |          |               |
> |               |    Uncal       |     0.2991$_{2.4}$      |     0.0608$_{4.1}$    |        0.1725$_{3.5}$        |
> |       Qsar-Biodeg        |      IR     |      0.2937$_{2.0}$     |      0.0604$_{3.2}$     |       0.1734$_{3.1}$        |
> |               |       RECAL    |     **0.2929$_{1.2}$**     |    **0.0504$_{3.6}$**     |        **0.1674$_{3.7}$**      |
> |               |           |           |          |               |
> |               |     Uncal      |      0.1739$_{2.8}$     |    0.0299$_{4.2}$     |     0.0899$_{0.9}$           |
> |         Seismic-Bumps      |    IR       |     0.1728$_{2.6}$       |     0.0290$_{1.7}$      |        **0.0882$_{0.4}$**       |
> |               |      RECAL     |     **0.1724$_{0.5}$**       |    **0.0216$_{3.4}$**     |     0.0899$_{0.2}$          |
>
> **GBDT:**
>
> | Dataset           | Calibration         |    NLL(↓)       |    ECE(↓)    |   BS(↓)   |
> |-------------------|---------------------|-----------------|---------------|----------------|
> |        | Uncal        |    0.2082$_{2.3}$    |    0.0159$_{3.7}$   |   0.1281$_{2.2}$ |
> |    Bank Marketing    | IR        |    0.2031$_{1.3 }$    |    0.0138$_{0.7}$   |   0.1279$_{2.1}$ |
> |        | RECAL        |   **0.2063$_{3.9}$**    |    **0.0127$_{2.5}$**   |   **0.1272$_{4.0}$**  |
> |               |           |           |          |               |
> |               |    Uncal       |     0.3061$_{2.6}$      |     0.0933$_{4.9}$    |        0.1833$_{3.3}$        |
> |       Qsar-Biodeg        |     IR     |   0.2993$_{1.9}$     |      0.0647$_{2.9}$     |       0.1725$_{7.1}$        |
> |               |       RECAL    |     **0.2816$_{3.0}$**     |    **0.0502$_{5.4}$**     |      **0.1717$_{2.8}$**      |
> |               |           |           |          |               |
> |               |     Uncal      |      0.1711$_{3.3}$     |    0.0317$_{5.0}$     |     **0.0872$_{1.4}$**            |
> |         Seismic-Bumps      |    IR       |     0.1683$_{5.4}$       |     0.0277$_{2.6}$      |        0.0881$_{3.2}$      |
> |               |      RECAL     |     **0.1674$_{4.3}$**       |    **0.0243$_{1.1}$**     |     0.0873$_{0.1}$          |
>
> **Tab Transformer:**
>
> | Dataset           | Calibration         |    NLL(↓)       |    ECE(↓)    |   BS(↓)   |
> |-------------------|---------------------|-----------------|---------------|----------------|
> |        | Uncal        |    0.2119$_{1.4}$    |    0.0141$_{3.1}$   |   0.1391$_{3.1}$ |
> |    Bank Marketing    | IR        |    0.2117$_{2.5}$    |    0.0111$_{1.8}$   |   0.1390$_{2.5}$ |
> |        | RECAL        |   **0.2087$_{2.1}$**    |    **0.0075$_{2.3}$**   |   **0.1388$_{3.6}$**  |
> |               |           |           |          |               |
> |               |    Uncal       |     0.2895$_{5.2}$      |     0.0816$_{6.7}$    |        0.1652$_{1.9}$        |
> |       Qsar-Biodeg        |     IR     |   0.2845$_{3.5}$     |      0.0796$_{2.6}$     |       0.1602$_{0.7}$        |
> |               |       RECAL    |     **0.2861$_{5.7}$**     |    **0.0600$_{7.6}$**     |      **0.1520$_{2.1}$**      |
> |               |           |           |          |               |
> |               |     Uncal      |      0.1803$_{4.0}$     |    0.0399$_{4.8}$     |     0.0886$_{1.6}$           |
> |         Seismic-Bumps      |    IR       |     0.1642$_{5.2}$       |     0.0261$_{2.3}$      |        0.0866$_{1.5}$      |
> |               |      RECAL     |     **0.1638$_{4.6}$**       |    **0.0212$_{5.6}$**     |     **0.0857$_{1.9}$**          |
>
> Histogram Binning changes the ranking of the model's predictive probabilities, which is inconsistent with our goal of not changing the accuracy of the model itself. Platt Scaling is often used to calibrate binary classification tasks, and our baseline TS has been shown to be superior to Platt Scaling (Guo et al., 2017). We will also keep an eye on the latest developments in scaling-based calibration methods and compare them with RECAL on tabular data.
>
> References：
>
> Chuan Guo, Geoff Pleiss, Yu Sun, and Kilian Q Weinberger. 2017. On calibration of modern neural networks. In International conference on machine learning, pages 1321–1330. PMLR.

---

### Meta-Review · Area_Chair_CTLj · 2023-09-15

**Recommendation:** 3

**Metareview:**

This paper delves into the problem of sample-relation guided confidence calibration on tabular data and introduces the RECAL framework, a post-training confidence calibration approach utilizing graph neural networks to model sample relations.

The overall consensus among most PC members is that this paper demonstrates good soundness but elicits mixed feelings regarding its level of excitement. After thoroughly reviewing the paper and the accompanying reviews, several notable concerns have emerged. These include:
1. Lack of Comparisons: The paper would benefit from providing comparisons with existing approaches or techniques to contextualize its contributions better.
2. Reproducibility: It is better to provide the source code for better reproducibility.

In my view, while the reviewers generally acknowledge the soundness of this paper, its excitement appears somewhat ambiguous. Consequently, I recommend that the authors carefully address these concerns during their revisions to strengthen the overall quality and impact.

---

### Decision · Program_Chairs · 2023-10-07

**Decision:**

Accept-Findings

**Comment:**

This paper delves into the problem of sample-relation guided confidence calibration on tabular data and introduces the RECAL framework, a post-training confidence calibration approach utilizing graph neural networks to model sample relations.

The overall consensus among most PC members is that this paper demonstrates good soundness but elicits mixed feelings regarding its level of excitement. After thoroughly reviewing the paper and the accompanying reviews, several notable concerns have emerged. These include:
1. Lack of Comparisons: The paper would benefit from providing comparisons with existing approaches or techniques to contextualize its contributions better.
2. Reproducibility: It is better to provide the source code for better reproducibility.

In my view, while the reviewers generally acknowledge the soundness of this paper, its excitement appears somewhat ambiguous. Consequently, I recommend that the authors carefully address these concerns during their revisions to strengthen the overall quality and impact.